# Solid-state laser refrigeration of a composite semiconductor Yb:YLiF$_4$ optomechanical resonator

Anupum Pant[1], Xiaojing Xia[2], E. James Davis[3] & Peter J. Pauzauskie [1,4,5 ✉]

Photothermal heating represents a major constraint that limits the performance of many nanoscale optoelectronic and optomechanical devices including nanolasers, quantum opto-mechanical resonators, and integrated photonic circuits. Here, we demonstrate the direct laser refrigeration of a semiconductor optomechanical resonator >20 K below room temperature based on the emission of upconverted, anti-Stokes photoluminescence of trivalent ytterbium ions doped within a yttrium-lithium-fluoride (YLF) host crystal. Optically-refrigerating the lattice of a dielectric resonator has the potential to impact several fields including scanning probe microscopy, the sensing of weak forces, the measurement of atomic masses, and the development of radiation-balanced solid-state lasers. In addition, optically refrigerated resonators may be used in the future as a promising starting point to perform motional cooling for exploration of quantum effects at mesoscopic length scales, temperature control within integrated photonic devices, and solid-state laser refrigeration of quantum materials.

---

[1] Materials Science and Engineering Department, University of Washington, Seattle, WA 98195-2120, USA. [2] Molecular Engineering & Sciences Institute, University of Washington, Seattle, WA 98195-1652, USA. [3] Chemical Engineering Department, University of Washington, Seattle, WA 98195-1750, USA. [4] Physical & Computational Sciences Directorate, Pacific Northwest National Laboratory, Richland, WA 99352, USA. [5] Institute for Nano-Engineered Systems, University of Washington, Seattle, WA 98195-1653, USA. ✉email: peterpz@uw.edu

Photothermal heating is a perennial challenge in the development of advanced optical devices at nanometer length scales given that a material's optical index of refraction, bandgap, and Young's modulus all vary with temperature. For instance, reducing the mechanical motion of an optomechanical resonator to its quantum ground state requires that the temperature ($T$) must be much less than $h\nu/k_B$, where $\nu$ is the mode frequency, $h$ and $k_B$ are Planck and Boltzmann constants, respectively[1]. Critically, incident laser irradiances must be kept low enough to avoid photothermal heating of the resonator above cryogenic temperatures[1–5]. Here, we demonstrate an approach for the photothermal cooling of nanoscale optoelectronic devices through the emission of anti-Stokes photoluminescence. In particular, we used a micrometer-scale grain of 10% $Yb^{3+}$-doped $YLiF_4$ (Yb:YLF) located at the end of a semiconductor optomechanical resonator (CdS) to cool the resonator >20 K below room temperature following excitation with a continuous wave laser source with wavelength $\lambda_0 = 1020$ nm.

The idea of refrigerating metallic sodium vapors using anti-Stokes luminescence was first proposed by Pringsheim in 1929[6]. Following the development of the laser, Doppler cooling of metallic vapors led to the first observation of Bose–Einstein condensates in 1995[7]. The first experimental report of solid-state laser cooling came in 1995 using $Yb^{3+}$-doped ZBLANP glass (Yb:ZBLANP)[8]. Since then, two decades of research in the area of solid-state laser refrigeration has culminated in the development of a solid-state optical cryo-cooler with bulk Yb:YLF single crystals grown using the Czochralski method[9], which has cooled crystals to 91 K from room temperature. The primary advantage of using crystalline materials for solid-state laser cooling is the existence of well-defined crystal-field levels, which minimizes inhomogenous broadening of rare-earth (RE) absorption spectra. Recently, this has enabled the first experimental demonstrations of cold Brownian motion[10] since Einstein's seminal paper[11] on Brownian motion in 1905. The increased optical entropy of the blue-shifted photons makes this cooling cycle consistent with the second law of thermodynamics[12].

Recently, laser refrigeration of a macroscopic Czochralski grown Yb:YLF crystal was used to cool a semiconductor FTIR detector (HgCdTe) to 135 K[13]. In contrast, in this work the temperature of a nanoscale semiconductor optomechanical resonator (CdSNR) is reduced using laser refrigeration of a hydrothermally synthesized Yb:YLF microcrystal attached to it. The cooling of a load using a microscopic cooler enables local cooling. In addition, it also offers a route towards rapidly achieving a thermal steady (μs to ms scale) temperature state within nanoscale devices.

The device is suspended in vacuum from a silicon wafer to reduce the potential for photothermal heating of the adjacent silicon substrate. Van der Waals bonding is used to attach a low-cost, hydrothermal ceramic Yb:YLF microcrystal[10] to the end of the resonator cavity. RE ($Yb^{3+}$) point-defects within the YLF emit anti-Stokes photoluminescence, which cools both the YLF microcrystal, and also the underlying semiconductor optomechanical resonator. The YLF serves both as a local thermometer (discussed in more detail below) and also as a heat sink, which extracts thermal energy from the cantilever, increasing its Young's modulus, and thereby blue-shifting the cantilever's optomechanical eigenfrequency. The transmitted laser causes minimal heating of the cantilever, supporting the YLF crystal owing to its small thickness (150 nm) and extremely low absorption coefficient of CdS at 1020 nm[14]. The temperature of the source and the cantilever system are measured using two independent non-contact temperature measurement methods—differential luminescence thermometry[15] and optomechanical thermometry[16], respectively, which agree well with each other.

The results below suggest several potential applications for using solid-state laser refrigeration to rapidly cool a wide range of materials used in scanning probe microscopy[17–19], cavity optomechanics[2,5,20,21], integrated photonics[22–24], the sensing of small masses and weak forces[25–28], quantum information science[29], and radiation-balanced lasers[30].

## Results

**Optomechanical thermometry**. A CdSNR was placed at the end of a clean silicon substrate, and a hydrothermally grown 10% Yb:YLF crystal was placed at the free end of the CdSNR cantilever. CdS was chosen because of its wide bandgap and low-cost, though in principle any material with low near-infrared (NIR) absorption can be used. A bright-field optical image of a representative sample is shown in Fig. 1b. The silicon substrate was loaded inside a cryostat chamber such that the free end of the cantilever was suspended over the axial hole in the cryostat, and the system was pumped to ~$10^{-4}$ Torr. As shown in Fig. 1a, a 1020-nm laser was focused onto the Yb:YLF crystal at the end of the cantilever. The time-dependent intensity of the forward-scattered 1020 nm laser was measured by focusing it onto an avalanche photodiode (APD). To measure the cantilever's eigenfrequencies the voltage vs. time signal was Fourier-transformed to obtain its thermomechanical noise spectrum[31,32]. A representative power spectrum measured on the sample at 300 K using a laser irradiance of 0.04 MW cm$^{-2}$ is shown in Fig. 1c. A sharp peak, fitted using a standard Lorentzian with a peak position at 3648.9 Hz, was attributed to the first natural resonant frequency mode ("diving board mode") of the fluorescence-cooled nanoribbon (FCNR) system (Supplementary Fig. 1). As shown in Fig. 1a, the backscattered photoluminescence was collected from the rear end of the objective, transmitted through a beamsplitter, was filtered using a 1000-nm short-pass filter and focused into the spectrometer slit. Photoluminescence (PL) spectra were recorded at different grating positions, with appropriate collection times to avoid saturating the detector, and they were stitched together. Ten spectra were collected using 0.04 MW cm$^{-2}$ of laser irradiance for 0.1 s and averaged. The intense $Yb^{3+}$ transitions[33,34] in the range of 800–1000 nm, with major peaks at 960 ($E_6–E_1$), 972 ($E_5–E_1$), and 993 ($E_5–E_3$) nm were observed (Fig. 1d). A longer acquisition time (50×) was used to collect the weaker luminescence signal from the other rare-earth (RE) impurities that were not explicitly added during synthesis. The up-converted green and red emission peaks at 520, 550, and 650 nm are attributed to the transitions from $^2H_{11/2}$, $^4S_{3/2}$ and $^4F_{9/2}$, respectively, of trivalent erbium ions ($Er^{3+}$)[35,36]. Other minor transitions are labeled.

Power spectra normalized using the maximum value at different laser irradiances obtained from the sample are plotted in Fig. 2a. When fit to a standard Lorentzian, the peak values show a blue-shift in the eigenfrequency of the FCNR system as the laser power is increased. The fitted peak values of these power spectra are shown in Fig. 2b. As the laser irradiance was increased, the Yb:YLF source reached lower temperatures, thereby extracting more heat from the CdSNR cantilever and causing a blue-shift in the frequency owing to an increased Young's modulus of the CdS at lower temperatures. Using a 980 nm laser with an irradiance of 0.5 MW cm$^{-2}$ resulted in the irreversible photothermal melting of the cantilever device shown in Supplementary Fig. 2. When the Yb:YLF crystal was removed from the CdSNR cantilever in Fig. 2b, the fundamental frequency measured at 0.04 MW cm$^{-2}$ increased to a higher value of 17384.3 Hz owing to the removal of mass from the system (~$1.3 \times 10^{-9}$ g, Supplementary Fig. 3). As a control experiment, the eigenfrequency of the CdSNR cantilever itself was measured

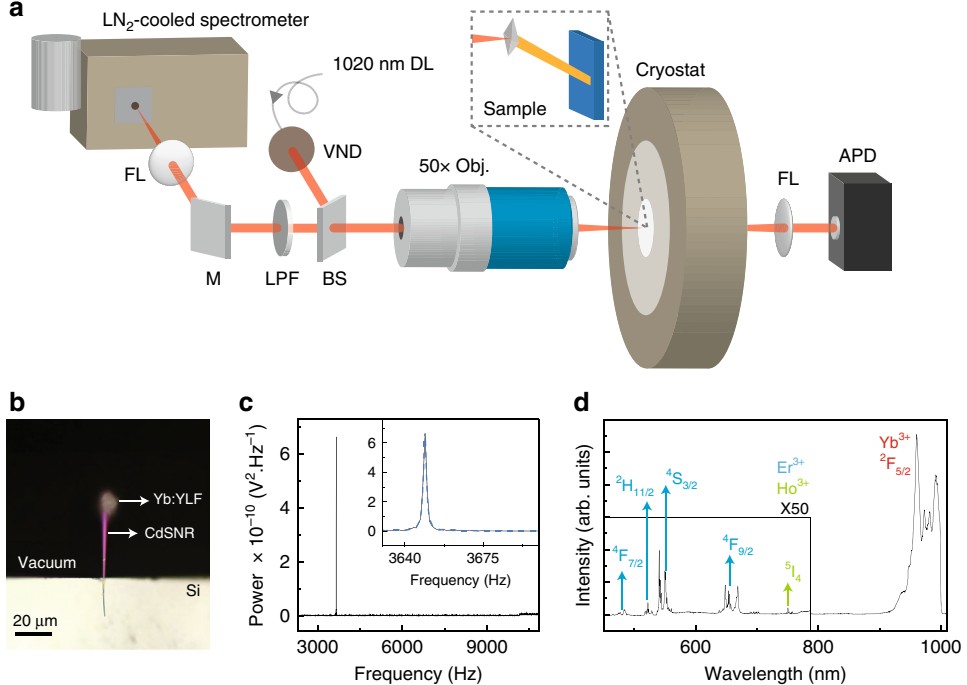

**Fig. 1 Experimental setup. a** The schematic of the eigenfrequency and up-converted fluorescence measurement setup. FL, M, SPF, DL, BS, VND, and APD stand for focusing lens, mirror, 1000 nm short-pass filter, diode laser, beamsplitter, variable neutral density filter, and avalanche photodiode, respectively. **b** A bright-field optical image of the CdSNR cantilever supported using a silicon substrate with a $Yb^{3+}$:YLF crystal placed at the free end. **c** A peak in the thermomechanical noise spectrum originating from the fundamental eigenfrequency of the CdSNR with $Yb^{3+}$:YLF sample obtained at the $0.04\,MW\,cm^{-2}$ at 300 K. **d** A stitched, up-converted fluorescence spectrum obtained at room temperature using a 1020-nm excitation source ($0.04\,MW\,cm^{-2}$) focused on the suspended $Yb^{3+}$:YLF crystal. A 1000-nm short-pass filter was used to cut off the laser line.

after the removal of the Yb:YLF crystal. The eigenfrequency of the cantilever without the crystal was then measured as a function of the laser power and is shown in Fig. 2b. The eigenfrequency red-shifted as the laser irradiance was increased, suggesting greater heating of the cantilever at higher irradiances owing to the decreasing Young's modulus at higher temperatures[16]. The temperature of the FCNR device was calibrated by increasing the temperature of the cryostat from 160 to 300 K, which showed a linear red-shift in the eigenfrequency of the cantilever. The slope of $-0.389\,Hz\,K^{-1}$ obtained using this calibration was used to measure the temperature change of the cantilever system during laser refrigeration experiments. The maximum blue-shift in the eigenfrequency as a function of laser irradiance of the was $+20.6\,Hz$ at an irradiance of $0.97\,MW\,cm^{-2}$, compared with the lowest irradiance of $0.04\,MW\,cm^{-2}$. Based on the isothermal temperature calibration, ignoring temperature gradients and other optomechanical effects on the cantilever owing to increased irradiance, this blue-shift of $+20.6\,Hz$ corresponds to a temperature change of 53 K below room temperature (assuming a negligible change in temperature at a laser irradiance of $0.04\,MW\,cm^{-2}$).

To evaluate the effect on the eigenfrequency at other wavelengths, we co-focused a 980 nm laser with the 1020 nm beam. The thermomechanical noise spectra of a representative device were measured using both wavelengths (Supplementary Fig. 4 and Note 1). Figure 2d shows that with increasing 1020 nm laser irradiance the eigenfrequency blue-shifts, indicating that the device cools. In contrast, using a 980 nm laser leads to a heating-induced red-shift of the cantilever's eigenfrequency with increasing irradiance. To limit damage (Supplementary Fig. 2) to the device we maintained the 980 nm irradiance below $0.2\,MW\,cm^{-2}$.

To obtain the absolute change in temperature it is important to consider the system using modified Euler–Bernoulli beam theory and include the effects of the laser-trapping forces[37] on the Yb:YLF crystal, which acts as a spring at the end of the cantilever (Supplementary Fig. 5). The eigenfrequency of the cantilever increases with increasing laser irradiance, owing to the increased optical spring constant. Analytically the eigenfrequency ($f_i$) in hertz, of a uniform rectangular beam is given by ref. [38]:

$$f_i = \frac{1}{2\pi}\frac{\Omega_i^2}{L^2}\sqrt{\frac{EI}{\rho}}. \tag{1}$$

Here $L$ is the length, $\rho$ is the linear density, $E$ is the Young's modulus, and $I$ is the area moment inertia of the cross-section of the beam. The $i$th eigenvalue of the non-dimensional frequency coefficient $\Omega_i$ satisfies the following equation for a uniform rectangular cantilever with a mass $M_0$ and spring of spring constant $K$ attached at the free end of the cantilever of mass $m_0$.

$$-\left(\frac{K}{\Omega_i^3} - \frac{\Omega_i M_0}{m_0}\right)[\cos(\Omega_i)\sinh(\Omega_i) - \sin(\Omega_i)\cos(\Omega_i)]$$
$$+ \cos(\Omega_i)\cosh(\Omega_i) + 1 = 0. \tag{2}$$

To experimentally probe the effects of the laser-trapping forces, the power-dependent eigenfrequency measurements were performed at a constant cryostat temperature of 77 K. At temperatures as low as 77 K, the cooling efficiency of the Yb:YLF crystal decreases owing to diminishing resonant absorption and red-shifting of the mean fluorescence wavelength[15]. Owing to negligible cooling with increased irradiance, and with the equilibrium temperature being maintained by the crysostat, it is assumed that any blue-shift in the eigenfrequency of the system was solely due to the greater laser-trapping force at higher

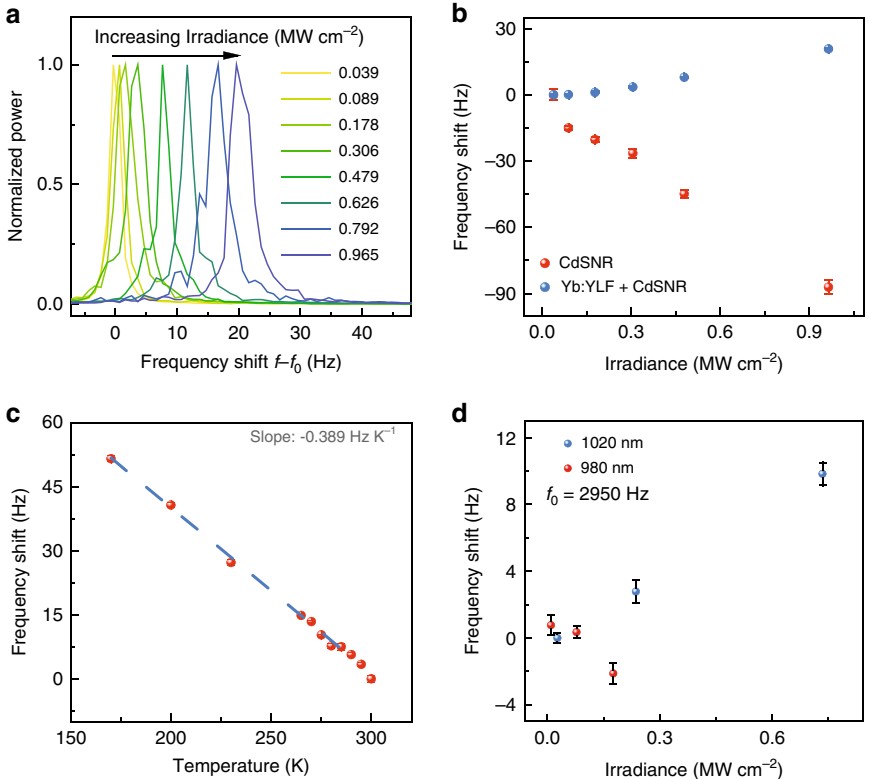

**Fig. 2 Eigenfrequency thermometry. a** Normalized power spectra for a representative laser refrigeration measurement at each laser irradiance with an ambient reference temperature of 295 K ($f_0 = 3632.2$ Hz). **b** The frequency shift ($f - f_0$) with laser power at 295 K for both a plain CdSNR (red) and CdSNR with Yb:YLF (blue). Each data point was obtained by taking the mean of peak position obtained from Lorentzian fits of six thermomechanical noise spectra and error bars represent one standard deviation. Note that for small standard deviations, the error bars overlap with the data point. $f_0$ is 3632.2 Hz and 17384.4 Hz, respectively. **c** Temperature calibration of the CdSNR with Yb:YLF obtained by measuring the frequency shift $f - f_0$ ($f_0 = 3653.6$ Hz) as a function of the cryostat temperature. The data points take into account the uncertainties in measurement by averaging the frequency value obtained from six thermomechanical noise spectra recorded at the given temperature. The error bars represent one standard deviation. **d** Eigenfrequency measurements using co-focused 980 nm and 1020 nm lasers. The data points take into account the uncertainties in measurement by averaging the frequency value obtained from six thermomechanical noise spectra recorded at the given laser power. The error bars represent one standard deviation.

irradiance (Supplementary Fig. 6). Therefore, the excessive blue-shift at room temperature ($6 \pm 2.2$ Hz) can be attributed to the change in Young's modulus due to cooling of the CdSNR cantilever (Supplementary Fig. 7). According to this calibration, the cantilever's temperature is reduced $15.4 \pm 5.6$ K below room temperature (Supplementary Fig. 8). As cantilever eigenfrequencies are calibrated at isothermal conditions, the temperatures measured via cantilever eigenfrequencies during laser refrigeration do not directly measure the coldest point within the cantilever, but rather a lower bound of the absolute minimum achievable temperature decrease[16]. This is a consequence of temperature gradients within the cantilever that lead to gradients of the cantilever's Young's modulus. Based on finite element eigenfrequency modeling of the cantilever with a spatially varying Young's modulus, the coldest point in the cantilever can be calculated (Supplementary Fig. 9). Below we present a steady-state, heat-transfer model of the laser-cooled cantilever system to quantify how thermal gradients within CdSNR cantilevers affect eigenfrequency measurements during laser cooling experiments.

**Heat-transfer analysis.** A cantilever of length 'L', width 'W', and thickness 'H' is modeled with a YLF crystal placed at the free end (see Fig. 3a). The YLF crystal is approximated as a cuboid with sides of $H_{YLF} = 6$, $L_{YLF} = 7.5$ and $W_{YLF} = 6$ μm, such that the volume and aspect ratio were similar to the tetragonal bi-pyramidal YLF crystal used experimentally.

At steady-state the temperature distribution in the nanoribbon satisfies the energy equation given by:

$$\frac{\partial^2 T}{\partial x^2} + \frac{\partial^2 T}{\partial y^2} + \frac{\partial^2 T}{\partial z^2} = 0. \tag{3}$$

Heat transfer to the surroundings by conduction and convection is absent owing to the vacuum surrounding the cantilever. Radiant (blackbody) energy transfer to or from the surroundings is negligible owing to the relatively low temperatures of the cantilever and its small surface area. Therefore, the heat flow within the cantilever is one-dimensional and Eq. (3) reduces to:

$$\frac{d^2 T}{dx^2} = 0, \tag{4}$$

which has the general solution:

$$T(x) = C_1 x + C_2. \tag{5}$$

Assuming negligible interfacial resistance between the cantilever and the underlying silicon substrate, the temperature at the silicon/CdS interface ($x = 0$) is the cryostat temperature $T_0$. Consequently, the boundary condition at the base of the nanoribbon is $T(0) = T_0$. If all of the heat generated in the YLF crystal is transferred to or from the CdSNR across the interface at $x = L_c$, the heat flux at the interface is given by:

$$\kappa \frac{dT}{dx}(L_c) = \frac{\dot{Q}_c}{HW}, \tag{6}$$

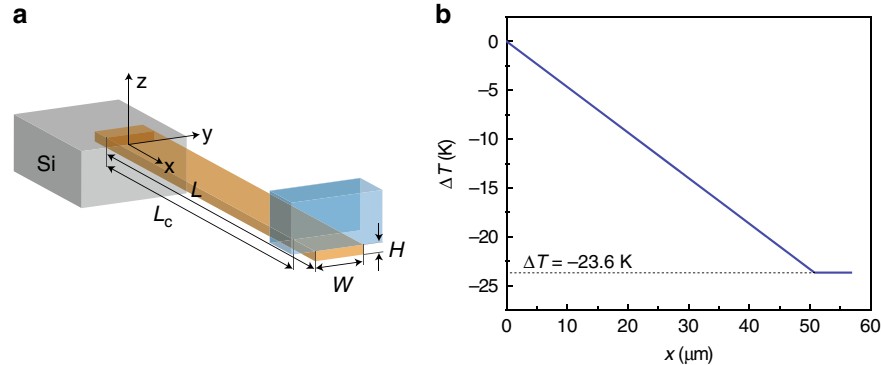

**Fig. 3 Heat-transfer analysis. a** The geometry of FCNR system used for analytical and finite element heat-transfer modeling. **b** The steady-state temperature along the length of the CdSNR calculated using analytical one-dimensional solution obtained assuming all of the cooling power produced by the YLF crystal flows through the CdSNR cross-section at $L_c$.

in which $\dot{Q}_c$ is the rate of heat removal from the YLF crystal, and $\kappa$ is the thermal conductivity of CdS.

Applying the boundary conditions, the temperature distribution in the CdSNR becomes:

$$T(x) = \frac{\dot{Q}_c}{\kappa HW}x + T_0. \quad (7)$$

It is assumed that the relatively large thermal conductivity of the YLF crystal ($\sim 6\,\mathrm{W\,m^{-1}\,K^{-1}}$) will lead to a nearly uniform temperature in the crystal given by:

$$T(L_c) = \frac{\dot{Q}_c}{\kappa HW}L_c + T_0. \quad (8)$$

The rate of laser energy absorbed per unit volume $Q''' = Q_{abs}/V$ is given by:

$$Q''' = \frac{4\pi n'n''}{\lambda_0 Z_0}(\mathbf{E}\cdot\mathbf{E}^*). \quad (9)$$

Here $n = n' - in''$ is the complex refractive index of the medium, $\lambda_0$ is the wavelength in vacuum, $Z_0$ is the free space impedance ($Z_0 = 376.73\,\Omega$), and $\mathbf{E}^*$ is the complex conjugate of the electric field vector within the YLF crystal. Up-converted, anti-Stokes luminescence follows laser absorption, cooling the crystal. We neglect the absorption of the incident laser by the underlying CdS cantilever due to its small thickness (154 nm) and low absorption coefficient at 1020 nm ($6.7 \times 10^{-13}\,\mathrm{cm^{-1}}$) relative to what has been reported[15] for YLF ($\sim 1\,\mathrm{cm^{-1}}$).

## Discussion

Given that eigenfrequency measurements can only provide a lower bound of the cantilever's temperature, a more direct approach must be used to measure the temperature at the end of the cantilever. Differential luminescence thermometry (DLT)[9,34] was used to measure the temperature of the YLF at the end of the cantilever based on using a Boltzmann distribution to analyze emission from different crystal-field (Stark) levels. We obtained a temperature drop of 23.6 K below room temperature ($\Delta T_{max}$) at an irradiance ($I_0$) of $0.97\,\mathrm{MW\,cm^{-2}}$ corresponding to an incident power $P_0 = 40.1\,\mathrm{mW}$ and spot radius $w_0 = 1.15\,\mu\mathrm{m}$ (Supplementary Fig. 10). Using the measured value of $T(L_c) - T_0 = 23.6\,\mathrm{K}$, $H = 150\,\mathrm{nm}$, $W = 2.5\,\mu\mathrm{m}$, $L_c = 53\,\mu\mathrm{m}$, and $\kappa = 20\,\mathrm{W\,m^{-1}\,K^{-1}}$[39], we calculate a cooling power of $\dot{Q}_c = 3.34 \times 10^{-6}\,\mathrm{W}$ (Supplementary Fig. 11). The resultant temperature gradient along the length of the device is shown in Fig. 3b. Based on the temperature gradient, by modeling a spatially varying Young's modulus, the coldest point in the cantilever from eigenfrequency measurements was calculated to be between 26 and 58 K below

room temperature (Supplementary Fig. 9). This agrees well with the coldest temperature measured using DLT.

An absorption coefficient and cooling efficiency of $0.61\,\mathrm{cm^{-1}}$ and 1.5%, respectively, have been reported previously for a bulk YLF crystal doped with 10% Yb-ions[15]. Based on this absorption coefficient, considering full illumination of the Yb:YLF crystal, a maximum cooling power of $2.2\,\mu\mathrm{W}$ would be generated when irradiated by a 40.1 mW pump laser. This cooling power is smaller than the experimental cooling power reported above. The discrepancy can be explained by two factors related to the symmetric morphology of the YLF microcrystals. First, the size of the YLF microcrystals is within the Mie-regime for light scattering and internal optical fields may be enhanced considerably owing to morphology dependent cavity resonances. Supplementary Fig. 12 presents two-dimensional finite-difference time-domain calculations showing that internal optical power within a YLF microcrystal can be twice as large as the incident power owing to internal cavity resonances. Consequently, first-order linear absorption calculations may underestimate the cooling power owing to an underestimation of the internal optical power of the pumping laser. Second, a combination of light-scattering and multiple internal reflections of the pump beam within the microcrystal can excite a larger volume of the crystal compared with the incident spot size. Supplementary Fig. 13 demonstrates that fluorescence is emitted throughout YLF microcrystal, including far from where the excitation laser is focused.

In conclusion, we demonstrate an approach to decrease the temperature of a nanoscale semiconductor optomechanical resonator by >20 K below room temperature using solid-state laser refrigeration of a Yb:YLF crystal. Thermometry and calibration of the fabricated device are performed using two independent methods—optomechanical eigenfrequencies and differential luminescence thermometry, respectively—which compare well with each other. A modified Euler–Bernoulli model is used to account for the laser-trapping forces, and the measured temperatures are validated using heat-transfer theory. A maximum drop in temperature of 23.6 K below room temperature was measured near the tip of the cantilever. Among other applications in scanning probe microscopy and exploration of quantum effects at mesoscopic length scales[28], optical refrigeration of a mechanical resonator could have significant implications for weak force and precision mass sensing applications[26,27], in the development of composite materials for radiation-balanced lasers[30], and local temperature control in integrated photonic devices[23,24]. In the future, solid-state laser refrigeration may also assist in the cooling of optomechanical devices by enabling the use of higher laser irradiances in the absence of detrimental laser heating.

## Methods

**Cadmium sulfide nanoribbon synthesis**. The CdSNRs were synthesized using a chemical vapor transport method discussed in a previous publication[16]. A precursor cadmium sulfide (CdS) powder in an alumina boat, which was placed at the center of a quartz tube. The silicon (100) substrates were prepared by dropcasting gold nanocrystals in chloroform. The precursor was heated to 840 °C. A carrier gas consisting of argon and 5% hydrogen was used to transport the evaporated species over to a growth substrate placed at the cooler upstream region near the edge of the furnace.

**Ytterbium-doped lithium yttrium fluoride synthesis**. The hydrothermal method used to synthesize single crystals of 10% ytterbium-doped lithium yttrium fluoride (Yb:YLF) nanocrystals was performed following modifications to Roder et al.[10]. Yttrium chloride ($YCl_3$) hexahydrate and ytterbium chloride hexahydrate ($YbCl_3$) were of 99.999% and 99.998% purity, respectively. Lithium fluoride (LiF), lithium hydroxide monohydrate (LiOH · $H_2O$), ammonium bifluoride ($NH_4HF_2$), and ethylenediaminetetraacetic acid (EDTA) were analytical grade and used directly in the synthesis without any purification. All chemicals were purchased from Sigma-Aldrich. For the synthesis of Yb:YLF, 0.585 g (2 mmol) of EDTA and 0.168 g (4 mmol) LiOH · $H_2O$ were dissolved in 10 mL Millipore DI water and heated to ~80 °C while stirring. After the EDTA was dissolved, 1.8 mL of 1.0 M $YCl_3$ and 0.2 mL of 1.0 M $YbCl_3$ were added and continually stirred for 1 hour. This mixture is denoted as solution A. Subsequently, 0.105 g (4 mmol) of LiF and 0.34 g (8 mmol) of $NH_4HF_2$ were separately dissolved in 5 mL Millipore DI water and heated to ~70 °C while stirring for 1 h. This solution is denoted as solution B. After stirring, solution B was then added dropwise into solution A while stirring to form a homogeneous white suspension. After 30 minutes, the combined mixture was then transferred to a 23 mL Teflon-lined autoclave (Parr 4747 Nickel Autoclave Teflon liner assembly) and heated to 180 °C for 72 h in an oven (Thermo Scientific Heratherm General Protocol Oven, 65 L). After the autoclave cooled naturally to room temperature, the Yb:YLF particles were sonicated and centrifuged at 4000 rpm with ethanol and Millipore DI water three times. The final white powder was then dried at 60 °C for 12 hours followed by calcination at 300 °C for 2 hours in a Lindberg blue furnace inside a quartz tube.

**Device fabrication**. Using a tungsten dissecting probe (World Precision Instruments) with a sufficiently small tip radius-of-curvature (<1 μm), mounted on to a nano-manipulator (Märzhäuser-Wetzlär), the CdSNRs were picked up and placed at the edge of a clean silicon substrate. Yb:YLF crystal was then placed at the free end of the cantilever using the same process.

**Eigenfrequency measurement**. The optomechanical thermometry setup consists of a 1020 nm diode laser (QPhotonics) focused, using a 50× long working distance objective, onto the sample placed inside a cryostat (Janis ST500) modified by drilling an axial hole through the sample stage. The forward-scattered light was collected through the axial hole and was focused onto an APD (Thorlabs APD430A). The time-domain voltage signal from the APD was then Fourier-transformed to obtain the thermomechanical noise spectrum with characteristic peaks from the fundamental eigenfrequency modes of the cantilever. For temperature calibration, the thermomechanical noise spectrum was recorded by varying the cryostat temperature and fitting the resulting peaks using a standard Lorentzian. Each data point represents the average of six measurements and the error bars represent the standard deviation.

## Data availability

The data that support the findings of this study are available from the corresponding authors on reasonable request.

## Code availability

Code or algorithm used to generate results in this study are available from the corresponding authors on reasonable request.

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

## Acknowledgements

A.P., X.X., and P.J.P. gratefully acknowledge financial support from the MURI:MARBLe project under the auspices of the Air Force Office of Scientific Research (award no. FA9550-16-1-0362). Sample characterization was conducted at the University of Washington Molecular Analysis Facility, which is supported in part by the National Science Foundation (grant no. ECC-1542101), the University of Washington, the Molecular Engineering & Sciences Institute, the Clean Energy Institute, and the National Institutes of Health. The authors gratefully acknowledge support from UW's Institute for Nano-engineered Systems.

## Author contributions

The Yb:YLF microcrystal materials were synthesized by X.X. The heat-transfer analysis was developed by E.J.D., A.P., and P.J.P. A.P. performed the experiments and also prepared final figures. All co-authors contributed to the writing and editing of the manuscript.

## Competing interests

The authors declare no competing interests.
