## [Peer Review File · Nature Communications]

Response to the reviewer:

Reviewer #1 (Remarks to the Author):

Thank the authors for the detailed reply. I do not have any questions.

Response: We thank the reviewer for their positive remark towards our revised manuscript.

Reviewer #3 (Remarks to the Author):

I went carefully through the new version of the manuscript and the response letter. I would like to thank the authors for carefully addressing my comments, especially in: repeating the experiment with the heating wavelength now demonstrating cooling much more convincingly(!); changing the title to a more physically relevant one and changing the focus of the motivation paragraph. In my opinion, the manuscript has improved both in the context and delivery and I recommend it for publication without hesitation.

Response: We thank the reviewer for their encouraging comments.

One thing I would like to bring up to the authors for their consideration is the possibility to change the way Fig 2D is plotted. At present, datapoints at 980nm appear asymmetric with respect to those obtained at 1020nm. On the other hand, if one would plot the abscissa in terms of the absorbed power, then the two datasets will be roughly in the same range of the x-axis, likely aiding the presentation and focusing more on the estimates which were carried out in response to Ref 1.

Response: We agree that the value of absorption plotted in the abscissa would help the datasets span a similar range. However, we think retaining the incident laser power may be more helpful for the reader to replicate the experiment as it can be measured and controlled far more easily, especially in the case of microcrystals. In addition, it helps us in avoiding the use of absorption values of bulk 10%Yb:YLF from the literature, as we do not measure the absolute absorption by individual microcrystals in this study.

Reviewer #1 (Remarks to the Author):

Thank the authors for the detailed reply. I do not have any questions.

Reviewer #3 (Remarks to the Author):

I went carefully through the new version of the manuscript and the response letter. I would like to thank the authors for carefully addressing my comments, especially in: repeating the experiment with the heating wavelength now demonstrating cooling much more convincingly(!); changing the title to a more physically relevant one and changing the focus of the motivation paragraph. In my opinion, the manuscript has improved both in the context and delivery and I recommend it for publication without hesitation.

One thing I would like to bring up to the authors for their consideration is the possibility to change the way Fig 2D is plotted. At present, datapoints at 980nm appear asymmetric with respect to those obtained at 1020nm. On the other hand, if one would plot the abscissa in terms of the absorbed power, then the two datasets will be roughly in the same range of the x-axis, likely aiding the presentation and focusing more on the estimates which were carried out in response to Ref 1.

Response to the reviewer:

Reviewer #1 (Remarks to the Author):

I appreciate the authors for the detailed reply. The response is very exciting and could give audience a better understanding on the field. I agree that my previous calculation cannot capture the effects of multi-reflection and reabsorption effect.

Response: We thank the reviewer for their positive remarks.

Here I modify my model. Since the pump intensity is pretty high, we consider the absorption saturation effect in the crystal.

For a crystal of 6 μm by 7.5 μm by 6 μm , assuming the crystal is fully illuminated, the total absorbed power is

$$\begin{aligned} P_{\text{abs}} &= \int I(x,y,z) * \alpha(I) dV \\ &= \int I * \alpha_0 / (1 + I/I_s) dV \\ &= \alpha_0 * I_s * \int (1 - 1/(1 + I/I_s)) dV \\ &\leq \alpha_0 * I_s * V < 0.6/\text{cm} * 100\text{kW}/\text{cm}^2 * (6 \mu\text{m} * 7.5 \mu\text{m} * 6 \mu\text{m}) = 16.2 \mu\text{W}. \end{aligned}$$

I could not find a good reference on the saturation intensity at 1020 nm for Yb:YLF. Based on studies with Yb:YAG, I expect it is in the range of 10-20 kW/cm². 100 kW/cm² is definitely an overestimation. If the previous calculation is correct, with 1.5% cooling efficiency, the cooling power is no more than 0.243 μW , which is much lower than the 3.34 μW in the analysis.

Response: We appreciate the insight from the new calculation which considers a fully illuminated crystal. In this manuscript we observe cooling at the maximum irradiance of 0.9 MW/cm² allowed by the present setup. According to the new calculation the highest laser irradiance used in the experiment (0.9 MW/cm²) would result in 145.8 μW of absorbed power. If 1.5% cooling efficiency is considered, this amounts to 2.2 μW of cooling power generated within the microcrystal. This agrees with the measured 3.34 μW , considering the uncertainties in the measurement of crystal dimension and the usage of a literature value for the cooling efficiency.

The cooling trend appears to continue beyond 0.9 MW/cm² and does not indicate saturation. We agree that the irradiances are higher when compared to macroscopic experimental setups which use Brewster's cut bulk Czochralski grown crystals irradiated with unfocused beams. However in previous studies at microscale using focused beams, Rahman *et al.* reported irradiances in the range of 8-35 MW/cm² to achieve lowest temperatures (up to 130 K) within micron scale 10%Yb:YLF particles[1]. Additionally, Roder *et al.* demonstrated that 10%Yb:YLF crystals are able to continue the cooling trend in a irradiance range of 5-25 MW/cm²[2]. Complexities in the microscopic environment such as the scattering losses, reflection losses, and changing waist radius over the crystal depth, leading to a changing irradiance of the focused Gaussian beam require a more detailed study to provide a more comprehensive model.

Accordingly, the following discussion on page 14 was modified using the new insights. It reads as follows:

“An absorption coefficient and cooling efficiency of 0.61 cm^{-1} and 1.5%, respectively, have been reported previously for a bulk YLF crystal doped with 10% Yb-ions.²⁶ Based on this absorption coefficient, considering full illumination of the Yb:YLF crystal, a maximum cooling power of $2.2 \text{ }\mu\text{W}$ would be generated when irradiated by a 40.1 mW pump laser. This cooling power is smaller than the experimental cooling power reported above. The discrepancy can be explained by two factors related to the symmetric morphology of the YLF microcrystals. First, the size of the YLF microcrystals is within the Mie-regime for light scattering and internal optical fields may be enhanced considerably due to morphology dependent cavity resonances. Figure S11 presents two-dimensional finite-difference time-domain calculations showing that internal optical power within a YLF microcrystal can be twice as large as the incident power due to internal cavity resonances. Consequently, first-order linear absorption calculations may underestimate the cooling power due to an underestimation of the internal optical power of the pumping laser. Second, a combination of light-scattering and multiple internal reflections of the pump beam within the microcrystal can excite a larger volume of the crystal compared to the incident spot size. Figure S10 demonstrates that fluorescence is emitted throughout YLF microcrystal, including far from where the excitation laser is focused.”

There are a few places the authors use kW/cm^2 and MW/cm^2 even though the absolute power level is quite close. It would be nice to be consistent.

Response: We have adjusted all the units in kW/cm^2 to MW/cm^2 to keep the units consistent throughout the manuscript.

Reviewer #3 (Remarks to the Author):

I thank the authors for taking their time to answer all of the questions posed by the referees. After reading through the reports, I agree with the second referee in that the experiments at the 'heating' wavelength would have been the most illustrative for understanding and proving whether the observable, in this case being the vibrational eigenfrequency of the cantilever, shows an opposite trend to that of cooling. I also understand the predicament the authors are in with the damaged sample, but I am also surprised at the claimed difficulty of reproduction of the samples. After all, the issue with the original cooling results reported for the CdS semiconductor laser cooling can be likely traced to sample non-reproducibility. I believe the authors should indicate in relevant places of the manuscript the difficulties associated with sample production and the issues/challenges which go along with its reproduction.

Response: We would like to thank the reviewer for their encouraging comments. In order to observe the opposite trend of the vibrational frequency caused due to the heating from a 980 nm laser we conducted new experiments on a newly made device. To prevent the device from permanent damage we use low laser irradiances of 980 nm laser and keep it below $0.2 \text{ MW}/\text{cm}^2$. As a result, figure 2 has been modified to include an additional panel (Fig. 2D) in which the new results have been shown. Correspondingly, the following discussion was added to the manuscript:

“To evaluate the effect on the eigenfrequency at other wavelengths, we co-focused a 980 nm laser with the 1020 nm beam. The eigenfrequency of a representative device (Fig. S13a) was measured using both wavelengths. Figure 2d shows that with increasing 1020 nm laser irradiance the eigenfrequency blue-shifts, indicating that the device cools. In contrast, using a 980nm laser leads to a heating-induced red-shift of the cantilever's eigenfrequency with increasing irradiance (Fig. 2d). To limit damage (Fig. S12) to the device we maintain the 980 nm irradiance below 0.2 MW/cm².”

Additional information regarding the optical image of the new device and thermomechanical noise spectra have been included in the supplementary information section 13.

The last point brings me to my old comment from round 1 which I believe the authors have misunderstood. I believe the motivation of the manuscript is unfocused at best, but more precisely is misleading. The authors focus on the case of CdS semiconductor on a cantilever and use currently-disputed Nature 2013 results, where cooling was reported, as their starting point and motivation. Then they argue that they can achieve laser refrigeration of a semiconductor, in fact, I quote directly from their reply to Reviewer 3: "Our work is important because it is highly reproducible and also demonstrates that optical refrigeration of a semiconductor nanostructure can also be used for applications in cavity optomechanics". This is quite sloppy use of technical language: "optical refrigeration of a semiconductor nanostructure" has not been achieved here! A more appropriate and technically precise description should be: "laser refrigeration of a Yb:YLF microcrystal resulted in lowering of the temperature of the semiconductor thermal load. I didn't find such slip-ups in the actual manuscript text, but I urge the authors to double-check the text carefully and in its entirety not to make clumsy and inaccurate claims.

Response: We agree that in the manuscript we do not directly cool the semiconductor resonator using a laser. Instead, we employ laser refrigeration of a Yb:YLF microcrystal to decrease the temperature of the semiconductor optomechanical resonator. In order to emphasize this difference, we have reviewed the manuscript to prevent the use of language that may have been unintentionally misleading.

We have modified the following paragraph on page 3 to clarify this difference.

“Recently, laser refrigeration of a macroscopic Czochralski grown Yb:YLF crystal was used to cool a semiconductor FTIR detector (HgCdTe) to 135 K²⁴. In contrast, in this work the temperature of a nanoscale semiconductor optomechanical resonator (CdSNR) is reduced using laser refrigeration of a hydrothermally synthesized Yb:YLF microcrystal attached to it. The cooling of a load using a microscopic cooler enables local cooling. In addition, it also offers a route towards achieving rapid thermal steady (μ s to ms scale) temperatures state within in nanoscale devices.”

In order to clarify the focus of the manuscript to the reader, we have also changed the title from.

“Solid-state laser refrigeration of a semiconductor optomechanical resonator”

to

“Solid-state laser refrigeration of a composite semiconductor / Yb:YLiF₄ optomechanical resonator”

The opening sentence of the concluding paragraph was modified. It now reads as follows:

“In summary, we demonstrate an approach to decrease the temperature of a nanoscale semiconductor optomechanical resonator by >20K below room temperature using solid state laser refrigeration of a Yb:YLF crystal attached to it.”

Furthermore, authors fail to site 2011 work from the University of New Mexico where they demonstrated laser cooling of a transparent semiconductor thermal load using optical refrigeration in Yb:YLF for the first time.

Response: We have added the reference (ref 24.) to this study in the revised version of our manuscript.

Finally, the CdS motivation used in lines 52-60 of the manuscript appears to be unnecessary: because (i) authors did not try to refute the claims of the Nature 2013 nor do they even extract any measurements/modeling results which address some of the claims of the CdS cooling paper; (ii) CdS is not a particularly great optomechanical material. The reference to Nature (2013) at the beginning then appears to be essentially pointless, as it is left hanging without any further discussion. I propose that the authors either think about how their measurements/results contribute to a particular point of view refuting/supporting the Nature 2013 results or they drop the reference all-together (or they find a logical and fine line or argument). This reference together with the sloppy language mentioned above could be mistaken for the claims of laser cooling of the semiconductor CdS! which is not the case here.

Response: We agree that the Nature 2013 reference may not be essential to motivate our results. Including the changes from the previous comments with insights from this comment, the opening line of the paragraph on page 3 reads:

“Recently, laser refrigeration of a macroscopic Czochralski grown Yb:YLF crystal was used to cool a semiconductor FTIR detector (HgCdTe) to 135 K²⁴. In contrast, in this work the temperature of a nanoscale semiconductor optomechanical resonator (CdSNR) is reduced using laser refrigeration of a hydrothermally synthesized Yb:YLF microcrystal attached to it. The cooling of a load using a microscopic cooler enables local cooling. In addition, it also offers a route towards achieving rapid thermal steady (μ s to ms scale) temperatures state within in nanoscale devices.”

For the rest, the authors have answered all of my questions to a satisfactory level. I believe the work is original and deserves to be published, however, the language of the paper has to be sharpened up and the motivation has to made much more clear and focused, compared to the present version.

Response: We thank the reviewer for recommending our manuscript for publication. Using insights from the comments, we have taken great care to polish the language and have made significant revisions to clarify the focus and scope in the revised manuscript.

References

- [1] A. T. M. A. Rahman and P. F. Barker, "Laser refrigeration, alignment and rotation of levitated Yb³⁺:YLF nanocrystals," *Nat. Photonics*, vol. 11, no. 10, pp. 634–638, 2017.
- [2] P. B. Roder, B. E. Smith, X. Zhou, M. J. Crane, and P. J. Pauzauskie, "Laser refrigeration of hydrothermal nanocrystals in physiological media," *Proc. Natl. Acad. Sci.*, vol. 112, no. 49, pp. 15024–15029, 2015.

Editorial Note: This manuscript has been previously reviewed at another journal that is not operating a transparent peer review scheme. This document only contains reviewer comments and rebuttal letters for versions considered at Nature Communications

Reviewers' comments:

Reviewer #1 (Remarks to the Author):

I appreciate the authors for the detailed reply. The response is very exciting and could give audience a better understanding on the field. I agree that my previous calculation cannot capture the effects of multi-reflection and reabsorption effect.

Here I modify my model. Since the pump intensity is pretty high, we consider the absorption saturation effect in the crystal.

For a crystal of 6 μm by 7.5 μm by 6 μm , assuming the crystal is fully illuminated, the total absorbed power is

$$\begin{aligned} P_{\text{abs}} &= \int I(x,y,z) * \alpha(I) dV \\ &= \int I * \alpha_0 / (1 + I/I_s) dV \\ &= \alpha_0 * I_s * \int (1 - 1/(1 + I/I_s)) dV \\ &\leq \alpha_0 * I_s * V < 0.6/\text{cm} * 100\text{kW}/\text{cm}^2 * (6 \mu\text{m} * 7.5 \mu\text{m} * 6 \mu\text{m}) = 16.2 \mu\text{W}. \end{aligned}$$

I could not find a good reference on the saturation intensity at 1020 nm for Yb:YLF. Based on studies with Yb:YAG, I expect it is in the range of 10-20 kW/cm². 100 kW/cm² is definitely an overestimation. If the previous calculation is correct, with 1.5% cooling efficiency, the cooling power is no more than 0.243 μW , which is much lower than the 3.34 μW in the analysis.

There are a few places the authors use kW/cm² and MW/cm² even though the absolute power level is quite close. It would be nice to be consistent.

Reviewer #3 (Remarks to the Author):

I thank the authors for taking their time to answer all of the questions posed by the referees. After reading through the reports, I agree with the second referee in that the experiments at the 'heating' wavelength would have been the most illustrative for understanding and proving whether the observable, in this case being the vibrational eigenfrequency of the cantilever, shows an opposite trend to that of cooling. I also understand the predicament the authors are in with the damaged sample, but I am also surprised at the claimed difficulty of reproduction of the samples. After all, the issue with the original cooling results reported for the CdS semiconductor laser cooling can be likely traced to sample non-reproducibility. I believe the authors should indicate in relevant places of the manuscript the difficulties associated with sample production and the issues/challenges which go along with its reproduction.

The last point brings me to my old comment from round 1 which I believe the authors have misunderstood. I believe the motivation of the manuscript is unfocused at best, but more precisely is misleading. The authors focus on the case of CdS semiconductor on a cantilever and use currently-disputed Nature 2013 results, where cooling was reported, as their starting point and motivation. Then they argue that they can achieve laser refrigeration of a semiconductor, in fact, I quote directly from their reply to Reviewer 3: "Our work is important because it is highly reproducible and also demonstrates that optical refrigeration of a semiconductor nanostructure can also be used for applications in cavity optomechanics". This is quite sloppy use of technical language: "optical refrigeration of a semiconductor nanostructure" has not been achieved here! A more appropriate and technically

precise description should be: "laser refrigeration of a Yb:YLF microcrystal resulted in lowering of the temperature of the semiconductor thermal load. I didn't find such slip-ups in the actual manuscript text, but I urge the authors to double-check the text carefully and in its entirety not to make clumsy and inaccurate claims. Furthermore, authors fail to cite 2011 work from the University of New Mexico where they demonstrated laser cooling of a transparent semiconductor thermal load using optical refrigeration in Yb:YLF for the first time. Finally, the CdS motivation used in lines 52-60 of the manuscript appears to be unnecessary: because (i) authors did not try to refute the claims of the Nature 2013 nor do they even extract any measurements/modeling results which address some of the claims of the CdS cooling paper; (ii) CdS is not a particularly great optomechanical material. The reference to Nature (2013) at the beginning then appears to be essentially pointless, as it is left hanging without any further discussion. I propose that the authors either think about how their measurements/results contribute to a particular point of view refuting/supporting the Nature 2013 results or they drop the reference all-together (or they find a logical and fine line or argument). This reference together with the sloppy language mentioned above could be mistaken for the claims of laser cooling of the semiconductor CdS! which is not the case here.

For the rest, the authors have answered all of my questions to a satisfactory level. I believe the work is original and deserves to be published, however, the language of the paper has to be sharpened up and the motivation has to be made much more clear and focused, compared to the present version.